# Adoption of Electricity in Rural Rwanda 10 Years after Connection

Lise Masselus [1,2] ✉, Jörg Ankel-Peters [1,2], Gabriel Gonzalez Sutil[3], Vijay Modi [3], Joel Mugyenyi [3], Anicet Munyehirwe[4], Nathan Williams [5] & Maximiliane Sievert [1]

Power grid extension into hitherto unconnected areas is a key policy goal in Sub-Saharan Africa. Yet, connection and usage rates remain low in rural grid-covered areas, at least in the short and medium run. This paper provides a long-term follow-up of a large grid extension program in rural Rwanda, analyzing electricity adoption over time in a panel of 41 communities electrified up to ten years ago. Using both survey and administrative data, we find that nearly half of the households in grid-covered communities remain unconnected. Even among those directly under the distribution grid, electrification rates barely exceed 80%. Electricity consumption and appliance use are low and have not increased over time. These findings suggest that, from an economic development or cost-benefit standpoint, rural grid investments are hard to justify. Instead, rights-based arguments centered on equity and fairness may offer a more compelling – albeit more controversial – justification for such investments.

Based on the belief that energy is a driver of economic development, universal access emerged as an important policy objective and a Sustainable Development Goal (SDG 7)[1]. Mission 300, a program launched in 2024 by the World Bank and the African Development Bank, for example, aims at connecting 300 million people in Sub-Saharan Africa (SSA) to lift them out of energy poverty and drive economic growth[2]. But the literature on the socio-economic impacts of rural electrification is divided, with a recent turn towards rather disappointing findings[3–7]. Impact evaluations of grid extension projects in SSA document connection rates well below 100%, very low consumption levels among those who are connected, and consequently, no impacts on economic development[8–13]. These studies typically examine adoption and impacts after two to three years, the longest after seven. Our paper responds to the legitimate criticism that adoption and impacts in those previous impact evaluations were measured too early, and that economic development effects might need more time to unfold[6,7,14]. Except for Burlig and Preonas[15], the few existing longer-run studies in the United States, Asia, and Latin America diagnose large effects[16–20]. Long-term evidence from the SSA context is not yet well documented.

The Rwandan Electricity Access Roll-out Program (EARP) under evaluation in this paper is one of the biggest grid expansion initiatives worldwide. EARP increased the household electrification rate from 6% in 2009 to 54% in 2023[21], making Rwanda the second fastest-electrifying country in SSA (after Kenya) in the last decade[22]. In a previous impact evaluation that was conducted 3.5 years after the first phase of EARP, Lenz et al.[11] document noteworthy usage patterns in households for lighting, studying, household chores, entertainment devices, and phone charging. Yet consumption levels were very low, also in connected enterprises, which hardly use electric appliances beyond lighting. Hence, there was no indication for the economic development effects of electrification that are typically assumed in cost-benefit considerations of donor agencies.

In the present paper, we provide a long-term follow-up on grid-based electrification in rural Rwanda, examining adoption of grid electricity in the same community sample as in Lenz et al.[11]—up to 10 years after they got access to the grid. By 2022, EARP successfully extended the grid to 41 communities and hence all but two communities of the Lenz et al.[11] sample. The average community in our sample was covered by the grid eight to nine years ago; 28 communities were

[1]RWI – Leibniz Institute for Economic Research, Essen 45128, Germany. [2]University of Passau, Passau 94032, Germany. [3]Columbia University, New York, NY, USA. [4]IB&C, Kigali, Rwanda. [5]Rochester Institute of Technology, Rochester, NY, USA. ✉e-mail: lise.masselus@rwi-essen.de

covered more than eight years ago and only four communities were covered less than five years ago. Our analysis is mainly based on four waves of self-collected survey data, but complemented by administrative consumption data and data from the Multi-Tier Framework (MTF). The survey dataset comprises detailed energy usage information from 820 households in 41 rural communities. The administrative consumption dataset includes all electricity purchases for 147,074 rural Rwandan households between 2012 and 2020, among them 174 households in our sample. To assess generalizability to the broader SSA context, we use data on electric appliance usage from MTF for five other SSA countries.

Since not all households in a community connect to the grid, throughout the paper we speak of communities being "covered" by the grid and of households being connected or non-connected. Using our survey data, we find that around half of households in grid-covered communities remain unconnected. Since distribution lines do not cover the whole community, we also examine connection rates among households living directly under the grid—so-called "under-grid" households henceforth. These households typically live within a 50 meter corridor on both sides of the low-voltage line and can connect at the lowest fee of 93 United States Dollars (USD) (we use the 2011 conversion rate of 600 Rwandan Francs (RWF) to 1 USD unless otherwise specified). Among under-grid households the electrification rate is higher at around 80%, but has stagnated since 2015. Electricity consumption among connected households remains very low, used mostly for lighting and phone charging. Only 23% of households own electric appliances other than a lamp, phone, or radio—mostly televisions. Few households use appliances for generating income or for productive purposes. Most enterprises acquire a connection, but there is no indication for noteworthy enterprise creation. These low consumption patterns are mirrored in administrative data on electricity purchases and appear consistent across other SSA countries using MTF data. Our findings suggest that, even in the long run, grid-based rural electrification in SSA may not yield the anticipated development impacts.

## Results

### Three datasets to triangulate adoption

Our analysis relies on three data sources. First, we collected four waves of a community panel dataset from 43 communities in 2011, 2013, 2015, and 2022 (the Methods section and Supplementary Note 1 provide more information on context, sample selection and the definition of a community). We dropped two non-covered communities from the Lenz et al.[11] sample because our community sample is too small to use them for further comparison. In all 41 covered communities, we used a random walk approach to select a sample representative of all under-grid households in the community. Per community, we interviewed 30 households in 2011 and 2013. A subsample was also interviewed in 2015. In 2022, we selected a new random sample of twenty under-grid households. Tracking the same households as in the previous waves was too complicated because family compositions, locations, and phone numbers changed. Also, for our research question, a household-level panel dimension is not important. Additionally, we interviewed community leaders to elicit information about the entire community, that is, including households living beyond the under-grid corridor as well as information about the enterprises and social infrastructure.

Our second dataset consists of administrative consumption data covering all purchases from October 2012 to April 2020 for 800,000 consumers of the national utility Rwanda Energy Group (REG, out of a universe of 1,300,000 consumers in total), from fifteen out of 30 districts in the country. Note that all connections in Rwanda have prepaid meters. We are interested in rural customers, but only about half of the consumers in the administrative data are geolocated, allowing us to identify them as rural or urban. From the geolocated data, we exclude urban households and only keep the 147,074 rural

households. We use this administrative dataset to scrutinize the external validity of our survey findings by comparing them to the rest of the country. The underlying assumption is that the administrative dataset is sufficiently representative despite the selection process, which is not entirely traceable. Furthermore, we use the administrative data to corroborate our survey consumption data. For this, we use a unique meter identifier to identify households that appear in both datasets. We could successfully match 26% of the connected households in our surveyed sample. In the Methods section, we implement a bias correction and find that the matched households are likely to represent the upper bound of the consumption distribution in our survey sample. The third dataset availed is the MTF data, collected between 2016 and 2018 by the World Bank. We use MTF datasets for Rwanda as well as Ethiopia, Niger, Nigeria, Kenya, and Zambia. We include all households with a grid connection in rural areas.

### Household adoption

We examine household consumption with survey data and the administrative consumption data. Figure 1a shows household connection rates and Fig. 1b shows appliance ownership over time from our survey data. Despite living in a covered community, not all households connect. In 2022, up to ten years after community coverage, 82% of the under-grid households are connected. At the entire community level, which includes households that live further away from the grid, only 51% are connected. Both observations are important from the universal access goal perspective. Especially in Rwanda's hilly terrain, extending the grid to the community center does not naturally imply that all parts of the community are within the reach of the distribution lines. But also the fact that the under-grid connection rate has hardly increased since our 3.5-year follow-up is very important. Those households who cannot afford the connection in the short term do not seem to save up money for a later connection. There is one unlikely but theoretically possible caveat: Between 2011 and 2022 REG might have extended distribution lines in some communities, and thus some of the 2022 under-grid households have not been under the grid for ten years. Among the 2011 under-grid households, the connection rate would then be higher than 82%. We cannot rule out that new distribution lines were added in some communities. Nevertheless, had lines been added systematically, it would have emerged in our community chief interviews, which it did not.

Adoption of electric appliances among connected under-grid households is low. In 2022, nearly all connected households use electric lamps, 84% own and charge mobile phones, and 50% own electric radios. Only 23% own other appliances (mostly televisions). Productive use of appliances is not common: only 4.5% of the connected households state that one or more of their electric appliances are used to earn money. These appliances are radios (10 respondents), irons (5), televisions (4), computers (3), refrigerators (2), a kettle (1) and a mill (1). They are all used for home business or farming activities. Figure 1b shows ownership of the most common appliances, mostly information and entertainment devices. There is no indication of increased appliance adoption over time.

In 2022, electric light bulbs have replaced kerosene lanterns and battery-run appliances almost entirely among connected households; only 14% use candles occasionally. Households not connected to the grid rely on battery-run Light-Emitting Diode (LED) torches, solar lamps, or candles. Like in most other countries in SSA, electricity is not used for cooking[23].

Figure 2 shows the self-reported average monthly consumption in kilowatt-hours (kWh). Figure 2a shows the 2013 consumption levels for connected households in those communities that were covered by 2013. Figure 2b shows consumption levels in 2022 for connected households in all communities. Up to ten years after electrification, the average connected household consumes 8.1 kWh per month, and the median is 4 kWh. Under current tariffs, the mean amount purchased

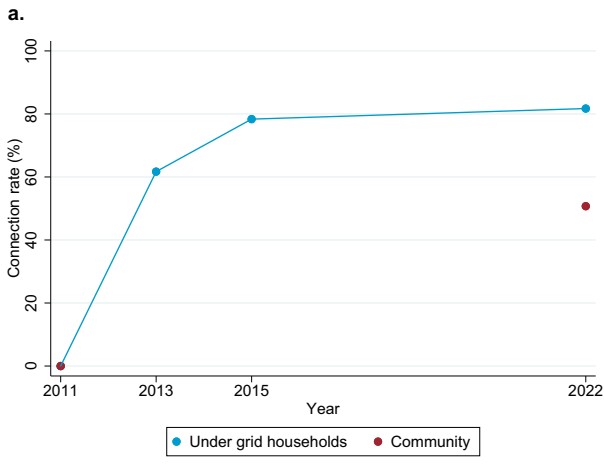

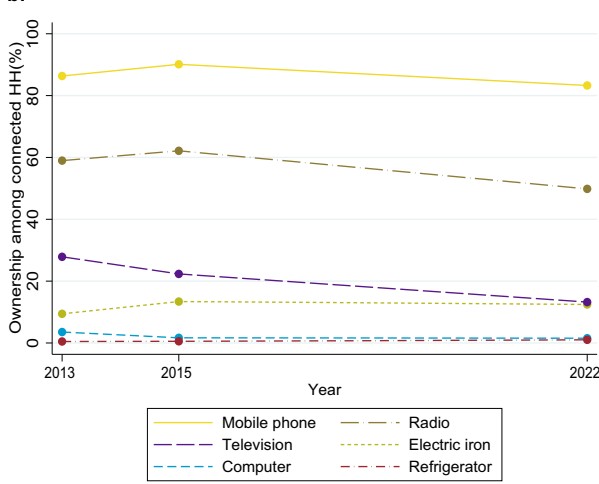

**Fig. 1 | Electricity adoption over time. a** shows the connection rates over time. A household is considered as grid-connected if they have a connection plus installation in their home, regardless of whether they consume any electricity. Following REG's definition, we do not count households that are connected through a neighbour. In contrast to other countries in the region, such illegal connections are rare in Rwanda. **b** shows appliance ownership over time. Data comes from the household and community surveys. The sample for both panels consists of thirteen covered communities in 2013 and 2015 and 41 covered communities in 2022.

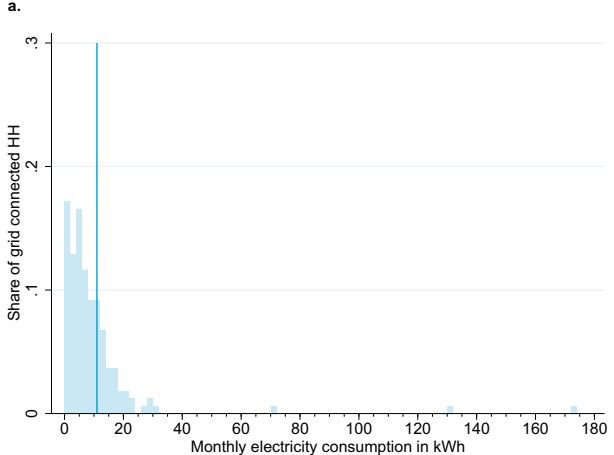

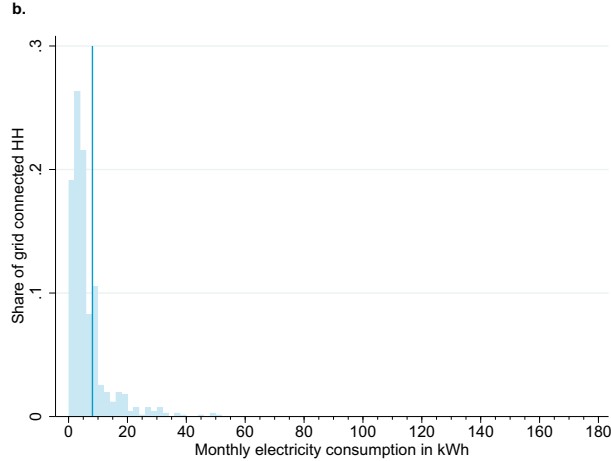

**Fig. 2 | Self-reported monthly electricity consumption for grid-connected households. a** shows the monthly electricity consumption for grid-connected households in 2013 and **b** in 2022. Data comes from the household survey. The sample for this figure consists of thirteen covered communities in 2013 **a** and 41 covered communities in 2022 **b**. When looking only at communities already covered by the grid in 2013, the mean consumption in 2022 is 8.5 kWh per month, indicating that the consumption in the entire sample is not pulled down by the more recently covered communities. Consumption is trimmed at the 99th percentile for graphical presentation. The vertical line presents the mean monthly consumption (untrimmed). The 2013 data is calculated based on appliance ownership and usage patterns. The 2022 data is based on electricity bills imputed with information on appliance ownership and usage. The approach is discussed in the Methods Section.

per month is 720 RWF, equivalent to 2% of the median household's expenditures.

Figure 3a shows the administrative consumption data for the 174 matched households. Earlier connected households initially have higher average levels of consumption, which could be explained by better-off communities being covered by the grid first, or better-off households within covered communities being connected first, or both. Additionally, we see no increase in consumption over time for any of the connection years. On the contrary, the data show a peak in consumption for the first years after electrification, which then tapers off. Both the frequency of purchases and the purchased quantity decline over time. We find similar trends for the highest 10% of consumers in our sample (see Supplementary Fig. 1), indicating that even for the top-consumers, there is no indication of consumption growth over time.

The administrative consumption data also corroborate our survey data results. The matched households have similar consumption levels in both the administrative consumption data and in the survey data, which gives confidence in the accuracy of the self-reported survey data (see the Methods section). Additionally, the administrative consumption data show that the low consumption levels in 2022 cannot be attributed to the COVID-19 pandemic, as consumption levels were already similarly low before the pandemic.

## Enterprise adoption and facilities

We elicited information from the community leaders about the enterprises in their community. In 2022, the average community has sixteen enterprises, most of them grid-connected, and all of which are micro-enterprises with no or very few employees. We find no evidence of substantial creation of new types of enterprises after electrification,

a.

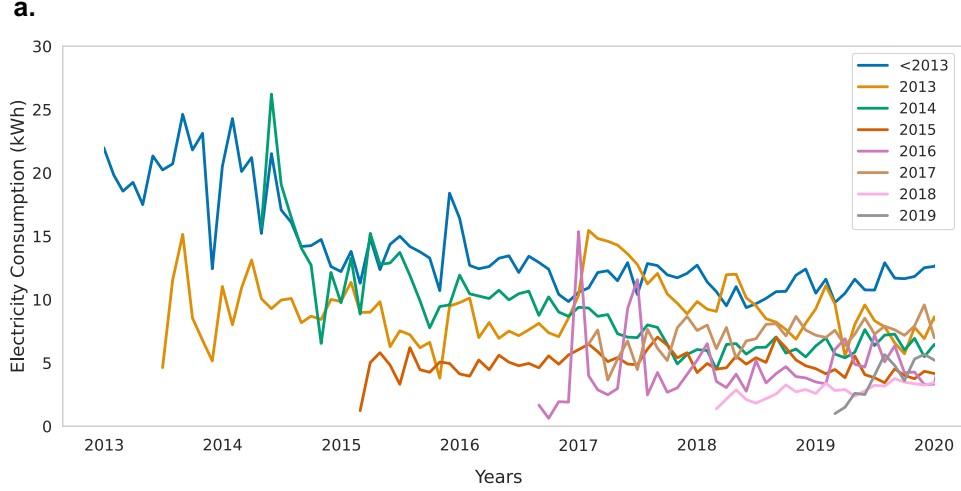

b.

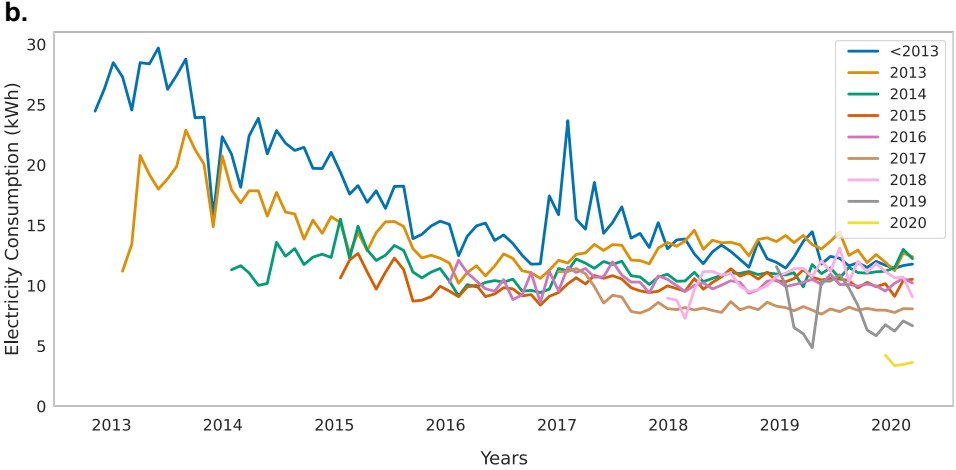

**Fig. 3 | Consumption over time by year of connection. a** shows the consumption over time for matched households in survey communities ($n = 174$). **b** shows rural households across the country ($n = 147,074$). Different lines represent the different years of connection. Data comes from the administrative consumption data which contains all pre-paid purchases and geolocations for 400,000 households. We use village boundaries provided by the World Bank and the definition of rural areas provided by the European Commission to identify rural customers.

although we do observe a few new enterprises that rely on electricity (like welders in 29% of the communities and copy shops in 12%). These enterprises might not have been created in the absence of grid-electricity because this would have required more costly power from a generator.

In general, enterprises in the communities provide basic services to the local population, like small shops, bars, restaurants, and hairdressers. Only half of the communities have any manufacturing firms like tailors, welders, or carpenters. Most enterprises use electricity for lighting. Small shops, selling items like staple food or toiletries for local consumption, as well as bars and restaurants sometimes obtain electric appliances after electrification. Among small shops, 22% have a radio, 12% a television, and 2% a refrigerator. Bars and restaurants often have a radio (50% and 22% respectively), or a television (35% and 22%). 17% of all bars and 14% of all restaurants have a refrigerator. Additionally, many shops, bars, and restaurants offer phone charging services. Hairdressers and beauty salons commonly own electric razors (93% of all hairdressers), a radio (44%) or a television (14%). Millers, carpenters, and tailors use grid electricity for operation of equipment, though many enterprises also continue to work with mechanical or diesel-run appliances, despite their grid

connection. Only 9% of all tailors have an electric sewing machine and 31% of all carpenters have some kind of electric wood processing machine.

We also asked community leaders about connection, appliance usage and noteworthy uptake of electricity at health centers and schools (although it was not the focus of our survey). First, it is important to observe that 30% of our communities do not have any schools or health infrastructure. Second, all fourteen secondary schools are connected, 21 of the 26 primary schools (81%) and ten of eleven health centers (91%). We furthermore understand that no new uses of electricity have emerged since 2013. The positive short-term results documented by Lenz et al.[11] regarding schools and health centers are energy cost savings, easier recruitment of skilled staff, improved administrative processes and the addition of some informatics classes for schools. While these impacts are likely to sustain beyond ten years after electrification, they are unlikely to be transformative with respect to poverty alleviation[24].

### Generalizability across Rwanda and Sub-Saharan Africa

In Fig. 3b, we compare consumption levels in our sample to the average in rural communities, using the administrative consumption dataset. The average consumption per rural household beyond our

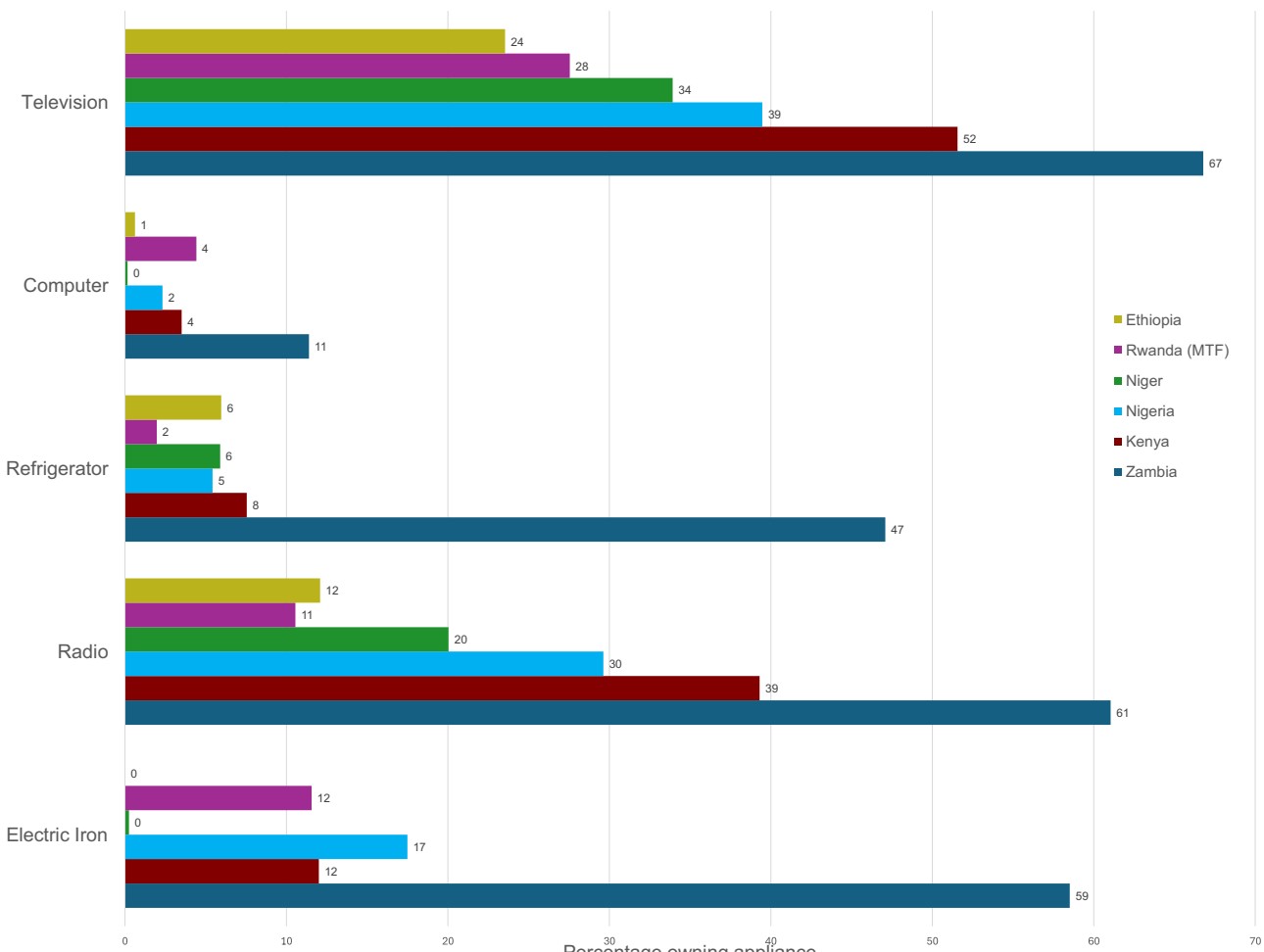

**Fig. 4 | Appliance ownership among rural grid-connected households across SSA.** This figure shows appliance ownership for common appliances for six countries. Data comes from the multi-tier framework survey. The sample consists of households with a grid connection in rural areas. All surveys were implemented between 2016 and 2018. The sample size is 309 households in Ethiopia, 618 in Rwanda, 817 in Niger, 530 in Nigeria, 521 in Kenya and 148 in Zambia. We dropped Liberia because of a mismatch between the main dataset and the appliance dataset.

sample is also low, and does not increase over time. Hence, our sample seems to be representative of the country with respect to adoption and consumption of electricity.

Next, we discuss the generalizability of our findings beyond rural Rwanda. We compare our estimates with connection rates and consumption levels from other studies in SSA, which mainly use utility data or nationally representative socio-economic datasets. Evidently, comparisons across contexts are difficult because of the variety in connection fees, tariffs, grid reliability, and other factors that may affect adoption. This section can therefore merely illustrate patterns observed across the continent. Blimpo & Cosgrove-Davies[25] report that connection rates for under-grid households vary substantially across SSA. Using Demographic and Health Survey data that includes urban areas, they approximate grid connection rates of between 13% and 81%. The median connection rate is 46%, with Rwanda slightly below average at 40%. No information is available for how long the areas have been electrified. In rural Kenya, Lee et al.[26] find that only 5% of households in their sample are connected up to five years after community connection. Half of the non-connected households live close to the grid and can connect at the lowest cost of 412 USD. In Tanzania, Bensch et al.[27] find that up to four years after connection, only 38% of households in grid-covered villages are connected despite high connection subsidies. Among under-grid households, the connection rate

is higher at 57%. Also in Tanzania, Chaplin et al.[9] document an increase in connection rates from 11 to 21% two to three years after a grid extension program. In Burkina Faso, Schmidt & Moradi[13] find that household connection rates stagnate at around 8–10% three years after community connection. In rural Ethiopia, 40% of households acquired a grid connection after 3 years[28]. These findings suggest that our diagnosis of low connection rates appears relevant for several SSA countries. The innovative insight we add is the long-term perspective.

On electricity consumption, Blimpo & Cosgrove-Davies[25] report an average of 483 kWh per year across SSA in 2014. This is equivalent to powering only a 50-Watt lighting source for a year, but is still four times the amount we measure in rural Rwanda. Descriptive studies using national utilities' data for Kenya[29,30] and Togo[31] document similarly low consumption and limited consumption growth over time. In Kenya, the median consumption among rural consumers ranges between 200-400 kWh per year in 2015. In Togo, the average consumption for rural consumers is around 600 kWh per year in 2020. In the first months after connection, electricity consumption grows modestly, but later tapers off. Similar to our findings in Rwanda, later connections consume less electricity.

Appliance adoption in rural SSA is also low. Figure 4 shows data from the MTF survey on appliance ownership among grid-connected households in rural areas. The data show that Rwanda is no outlier in

terms of appliance ownership. Using the 2015 exchange rate in order to make the tariff comparable to 2015 tariffs reported by Witte & Foster[32], Rwanda's lifeline tariff introduced in 2017 amounts to roughly 0.13 USD per kWh, and is thereby positioned moderately within the range of lifeline tariffs in Sub-Saharan Africa. While some countries do have lifeline tariffs as low as 0.01 USD per kWh (for example Ethiopia and Cameroon), many countries charge their lowest consumption customers more than 0.15 USD per kWh (Benin, Burkina Faso, Niger, Nigeria, Uganda). Figure 4 shows that countries with very low or high lifeline tariffs do not stand out in terms of appliance ownership, suggesting that tariffs are unlikely to be the main driver of appliance ownership. This is corroborated by Mugyenyi et al.[33] who analyze tariff changes in Rwanda and find that reducing the tariff by almost 50% increases electricity consumption only by slightly more than 10% for the lowest consumption tier and even less for the medium and highest consumption tiers.

## Discussion

We address two important policy questions: what is the electricity adoption trajectory over time, and will economic development impacts unfold in the long run? We do not observe a decline in connection rates and appliance usage over time. This, as well as the probably sustaining positive short-term impacts on household well-being observed in Lenz et al.[11], cannot be taken for granted in the aftermath of the COVID-19 pandemic. Yet, we also confirm Lenz et al.'s[11] short-term diagnosis of no noteworthy economic development effects, extending it to the long term. Moreover, we show that the universal access goal is not achieved automatically by rolling out the grid, with people staying unconnected even many years after connection, despite connection fees that are comparatively low in Rwanda[25,34]. Note that in contrast to some other countries in SSA, electricity supply is stable in rural Rwanda. Although blackouts and voltage fluctuations occur, they are infrequent (see Supplementary Note 1 for more details).

Our findings challenge the assumption, or hope, that development impacts unfold in the long run. Large donors and development banks conduct cost-benefit analyses to justify their investments, often based on very long-term amortization periods. World Bank's cost-benefit analyses, for example, project benefits over a 25 to 30-year time period[35]. Similarly, the Millenium Challenge Corporation employs a twenty-year horizon for its calculations[36]. Project evaluations typically look at time periods that are much shorter. Our findings ten years after the extension of the grid cast doubts on significant impacts materializing in the long run. Donors might nevertheless assume demand increases in the medium to long term, but such assumptions should be labelled as optimistic scenarios.

From this strict economic perspective our findings raise questions about whether grid extension should be the dominant strategy to reach the universal access goal. Investment requirements are indeed enormous in current scenarios with a dominant role for the grid. For SSA, the International Energy Agency (IEA) estimates that achieving universal access by 2030 and maintaining it to 2040 will cost over 100 billion USD per year. The annualized investments amount to 2.7% of the regional Gross Domestic Product (GDP)[37]. Alternatives to grid extension exist. For our sample of rural Rwandan households, most current electricity usage could be covered, for example, by decentralized off-grid home solar, which is availble at very low cost[38]. For the mini-grid sector, our findings are also a call for caution. Financial sustainability assessments in business plans are typically based on scenarios with increasing electricity demand over time—which often-times does not materialize[39–41].

Proponents of grid extension argue that in the long run it enables better development potential once economic growth and productive demand occur endogenously, whereas a focus on home solar would cap that demand increase. The same argument applies to cases where an exogenous stimulus accelerates growth in combination with electrification. Fetter & Usmani[5] observe that electrification led to economic development in communities in northwestern India that were simultaneously exposed to a positive exogenous price shock for agricultural products, but not in communities without this shock. While this demonstrates that electricity can enable development in certain places, it does not justify supplying electricity everywhere. The question hence arises as to whether such potential benefits, either in the far-away future or hinging on exogenous positive shocks, justify today's high investment and opportunity costs.

A careful case-by-case consideration of different electricity sources for different locations would improve the cost-effectiveness of scarce resources for universal access purposes[39,42–45]. Electrification interventions could focus on creating industrial zones in rural centers and towns. Alternatively, policy could target existing productive users in regions not covered by the grid, so-called anchor customers. This would be a version of the historical approach in today's high-income countries, where rural electrification typically followed rather than preceded industrial take-off[46].

Meanwhile, there are other reasons beyond economic ones that justify rural electrification. Many people and also governments in SSA perceive electricity and other infrastructure as a right that is derived from normative principles and not from cost-benefit considerations[47–49] (see also Supplementary Note 3). This also becomes manifest in countries where politicians use electrification to please (potential) voters[50]. From this rights-based vantage point, our findings are not disappointing since they confirm sustainable uptake of electricity—but, still, measures need to be taken to address stagnating connection rates and ensure all households under the grid are connected.

## Methods
### Sample selection

The communities in our sample are representative for communities scheduled to be electrified during EARP's first phase between 2009 and 2013. A representative random sample of then-treatment communities was chosen by Lenz et al.[11] according to probability-proportional-to-size sampling from a list of communities scheduled for electrification between 2011 and 2012. Then-control communities were partly selected from a list of EARP communities scheduled for connection after 2013, and partly according to their comparability to the treatment communities regarding road access, community size, number and type of enterprises, and prevailing agricultural activities.

The study communities are located across rural Rwanda. In comparison to other rural areas in SSA, rural Rwanda is relatively densely populated and could therefore represent an upper bound in terms of the potential benefits of rural electrification in SSA. We define a community as a group of households, clustered around basic infrastructure (see Supplementary Note 2). One community often covers multiple administrative settlements, referred to as imidugudu. The low-voltage lines often run in the center of the community, commonly alongside the main road.

In 2011, the average population is 300 households per community. All communities are located in rural areas, where the majority of the population relies on farming as their primary source of income. Only few communities had a business center or substantial entrepreneurial activity before grid connection. Existing enterprises mostly offered goods or services for local consumption, such as shops, hairdressers, bars, carpenters and tailors. The majority of communities are accessible via dirt road only, and only one is accessible via an asphalted road. Few communities have public facilities, apart from a primary school.

A random walk approach was used to select a sample of households representative of all under-grid households. Arriving in a community, the survey team first identified the 50 meters

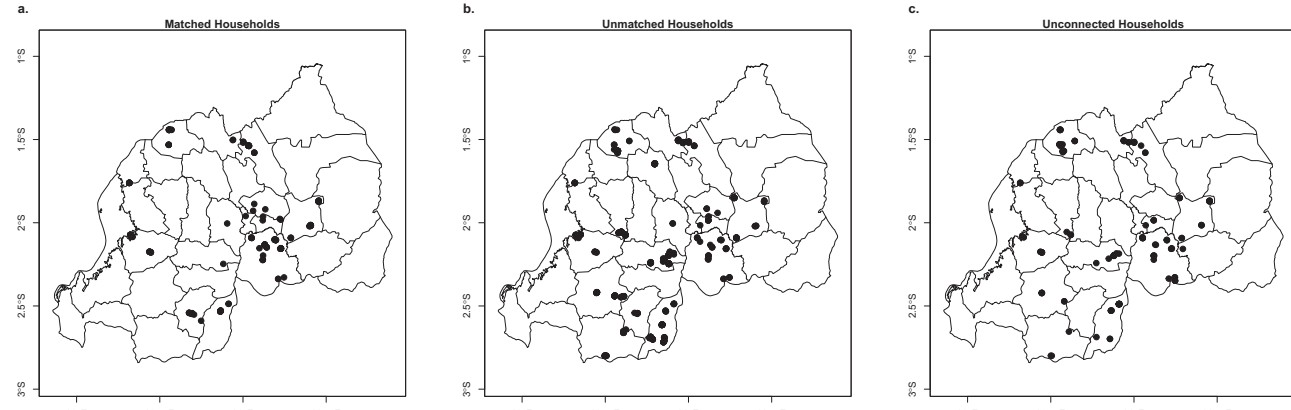

**Fig. 5 | Geographic distribution of matched and unmatched households. a** maps the households from our survey sample that are matched with the administrative consumption data. All matched households are connected to the grid. **b** maps the households from our survey sample that have a grid connection but are not matched with the administrative consumption data. **c** shows the households that are not connected to the grid.

corridor on both sides of the distribution lines. They then estimated the total number of households living in this corridor and interviewed every xᵗʰ household, with x being the total number of households in the corridor divided by the number of interviewed (30 in 2011 and 20 in 2022). A larger sample size in 2013 was required to test hypotheses with higher statistical power requirements (published in Lenz et al.[11]) than the questions addressed in this paper. The reduced sample size does not affect the representativeness of our sample, as a new random sample was drawn.

### Electricity consumption

Eliciting electricity consumption through recall in household surveys is challenging as households recharge their pre-paid meters on an as-needed basis and few households keep receipts. We elicit electricity consumption in two ways. First, we ask households for the amount consumed on the last three pre-paid bills (in kWh or RWF), the dates of recharge, and the average frequency of recharge. Second, we elicit ownership of appliances and lighting devices, and their average usage hours in each household. We use this data, and the average kWh per appliance, to infer monthly consumption.

Our preferred metric (as reported in Fig. 2) consists of a combination of these different variables. We use prepaid electricity bills for households that are able to provide them, and appliance and lighting usage for all other households. A total of 77 households are able to provide us with the exact date and recharge amount for at least two of their last three bills. For an additional 321 households, we use the average frequency of recharge. For the remaining 272 households, we only have data on appliance ownership, so we estimate electricity consumption based on appliance ownership and self-reported average usage hours.

The average consumption level for the different measures ranges between 6.1 kWh and 11.7 kWh per month. We assess the quality of our inferred values by comparing the values from bill dates and the inferred comsumption values for households for which we have two or more metrics. The Pearson correlation coefficients range between 0.14 and 0.48, which is moderate to high.

The administrative consumption data also show similar levels of consumption for the matched households. We compare the 2019 data from the administrative consumption data with our preferred combined consumption metric, described above. On average, the self-reported data from 2022 is higher than the administrative data from 2019, although the differences are minor. The average difference in electricity consumption for matched households in the two datasets is

4 kWh and the median difference is 1 kWh. The Pearson correlation coefficient is moderately high, at 0.29.

### Bias correction

We were able to match 26% of the connected households in our sample with the administrative consumption data. Figure 5 shows the distribution of households across the country.

Using our 2022 level survey data and a tobit model, we analyse which communities have the largest share of matched households. We find that the share of matched households is larger in communities with better road quality, a higher number of enterprises and in earlier electrified communities. In addition, at the household level, higher-income households with better housing quality are more likely to be matched. This means that the matching is imperfect and results in the systematic exclusion of some households.

To address this distorted matching, we formally implement a bias correction for the administrative consumption data using inverse probability weighting. We first estimate a probit regression which includes both household- and community-level covariates, such as demographic characteristics of the head of household (gender, age, education), household characteristics (number of members and their age), type of electricity access, dwelling characteristics (type of walls, windows, floors, and roofs), and community indicators. We use the fitted model to predict the probability to be matched, and then weight the observation by the inverse of the predicted probability.

Table 1 shows the descriptive statistics for yearly consumption from the administrative consumption data. The unweighted yearly mean and median is slightly higher than the yearly mean and median using inverse probability weighting, with 2014 and 2015 as exception. This indicates that the matched administrative consumption data used in the paper rather presents an upper bound for the entire survey sample.

### Inclusion & ethics in global research

We are committed to conducting ethical and inclusive research that respects the contributions, rights, and interests of all stakeholders. This study was carried out in close collaboration with local partners and institutions in Rwanda.

A co-author from IB&C, based in Rwanda, played a central role in the design, implementation, and interpretation of this research. Our work has been conducted in alignment with national policies and priorities, and we have actively engaged with relevant government authorities and institutional representatives throughout the research process.

**Table 1 | Unweighted and weighted yearly average consumption**

| Year | On-grid* | Not Weighted | | | | | Inverse Probability Weighting | | | | |
|---|---|---|---|---|---|---|---|---|---|---|---|
| | | Mean | Median | St. Dev. | Min. | Max. | Mean | Median | St. Dev. | Min. | Max. |
| 2013 | 47 | 148.37 | 72.20 | 196.97 | 0 | 968.00 | 142.85 | 52.20 | 194.25 | 0 | 968.00 |
| 2014 | 66 | 107.55 | 55.70 | 146.76 | 0 | 770.90 | 112.61 | 59.65 | 145.64 | 0 | 770.90 |
| 2015 | 82 | 89.09 | 40.80 | 123.61 | 0 | 600.60 | 94.70 | 45.65 | 121.50 | 0 | 600.60 |
| 2016 | 102 | 75.21 | 42.00 | 114.59 | 0 | 923.50 | 71.51 | 39.85 | 96.42 | 0 | 923.50 |
| 2017 | 107 | 84.14 | 47.50 | 138.33 | 0 | 1079.80 | 77.44 | 46.70 | 122.64 | 0 | 1079.80 |
| 2018 | 130 | 68.22 | 40.35 | 90.25 | 0 | 603.70 | 61.56 | 35.05 | 80.98 | 0 | 603.70 |
| 2019 | 165 | 58.39 | 32.40 | 76.80 | 0 | 610.00 | 52.21 | 27.25 | 69.14 | 0 | 610.00 |
| Total | 165 | 80.91 | 39.40 | 120.71 | 0 | 1079.80 | 74.69 | 36.35 | 108.83 | 0 | 1079.80 |

Source: Administrative consumption data. Values for consumption are in kWh. Nine households are excluded from this table as they connected during 2019/2020.
*On-grid are the number of meters at the beginning of the year.

We confirm that all required research permits, ethical clearances, and authorizations were obtained from the appropriate national and local authorities. We remain in regular dialogue with stakeholders to ensure ongoing transparency and mutual benefit.

This collaborative approach reflects our commitment to equitable partnerships, capacity sharing, and responsible conduct in global research.

### Ethical Compliance Statement

This study complies with all relevant ethical regulations. The research protocol was reviewed and approved by the Institutional Review Board, affiliated with the University of Connecticut (Protocol#: H20-0104), which provided oversight and guidance throughout the study. All study procedures were conducted in accordance with the ethical standards set forth by this committee. We also obtained clearance for data collection from the National Institute of Statistics of Rwanda (NISR). Informed consent was obtained from all participants prior to their involvement in the study.

### Reporting summary

Further information on research design is available in the Nature Portfolio Reporting Summary linked to this article.

## Data availability

The survey data used for creating Figs. 1 and 2 have been deposited on the OSF platform (https://doi.org/10.17605/OSF.IO/XQYST). The administrative data used in Figs. 3 and 5 cannot be shared due to privacy issues. Access might be granted after applying at info@reg.rw. Datasets used from the Multi-Tier Framework (MTF) Surveys used in Fig. 4 are provided by the World Bank Group and accessed via https://energydata.info/dataset. The core datasets for Rwanda, Kenya, Nigeria, Zambia, and Ethiopia were downloaded on September 6, 2021. The dataset for Niger was downloaded on March 29, 2021, and the dataset for Liberia was downloaded on September 20, 2021. Additional variables—specifically household weights and unique identifiers—for Ethiopia, Liberia, and Nigeria were not available at the time of original download and were subsequently retrieved on August 15, 2022. The datasets are available under the Creative Commons Attribution 4.0 and the CC0 1.0 license.

## Code availability

The code for creating Fig. 1 and Fig. 2 is available on the OSF platform (https://doi.org/10.17605/OSF.IO/XQYST).

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

## Acknowledgements

This research received funding from the JPAL – King Climate Action Initiative (K-CAI), grant GR-1786 and the International Growth Center, project code RWA-20187. We are grateful for comments received by participants at the Energy and Development seminar at Duke, the EAERE conference, the EfD annual conference and the LEADS conference. We thank Eugene Tan Perk Han and Courage Ekoh for providing input, as well as Erwin Bulte and Colin Vance for their very valuable suggestions. We thank REG for providing electrification data and IB&C Rwanda for their excellent work in collecting survey data.

## Author contributions

L.M., J.A.-P., A.M., and M.S. conceptualized the study. L.M., A.M., and M.S. collected the data. L.M., G.G.S., and J.M. performed the formal analysis. L.M. wrote the original draft. L.M., J.A.-P., M.S., and N.W. contributed to review and editing of the manuscript. J.A.-P., V.M., and N.W. supervised the project.

## Funding

## Competing interests

The authors declare no competing interests.
