## [Transparent Peer Review file · Nature Communications]

Adoption of Electricity in Rural Rwanda 10 Years after Connection

Corresponding Author: Dr Lise Masselus

Version 0:

Reviewer comments:

Reviewer #1

(Remarks to the Author)

Overall Assessment:

The paper adds to the existing body of knowledge on the question of whether grid extension leads to increased electricity access and usage in rural areas, using Rwanda as a case study. The uniqueness of the study lies in its long-term assessment (10 years) of electricity uptake in rural areas after grid extension compared to existing studies which have often focused on short to medium-term assessments. Despite this long-term assessment, the study does not report any significant new findings compared to what has already been reported by previous studies—connection rates, electricity consumption, and appliance use remain low in rural areas even after grid extension. Overall, the paper is well-structured and well-written, however, this reviewer finds its contribution to be quite limited to warrant its publication in Nature Communications.

Other issues

- 1) When the grid is extended to a community, it is considered “covered” not “connected” i.e electricity coverage vs electricity access (households connected). Authors should check and rectify this throughout the manuscript. E.g. It is stated in the abstract that “We find that in connected communities, almost half of the households remain unconnected.” This statement could be written as “We find that in covered communities, ...” Please check for similar terminology in other parts of the manuscript—under section 2.2. it is stated that “Despite living in a connected community, not all households connect.”
- 2) In other sub-Saharan African countries where elections are fiercely contested between opposition parties, grid extension is often a strategy for soliciting votes by incumbent parties during elections. In Ghana for example, it is not uncommon to see sign posts along roads in rural areas bearing the inscription “No electricity, no vote.” Hence, the impetus for grid extension may not necessarily be economic-driven or rights-based. But given that Rwanda is more or less a one-party state, this could hardly be the case and as such an economic or right-based impetus would hold true. A generalization of the results to other sub-Saharan Africa countries must take this third political factor into account.
- 3) For the surveys in 2022, why was the sample size reduced to 20?

(Remarks on code availability)

No code was provided

Reviewer #2

(Remarks to the Author)

The aim of the article is to assess whether access to electricity translates into actual long-term (10 years) household electricity usage. Its main contribution lies in adopting a long-term perspective, which is often lacking in similar studies, with Rwanda as a case study. Overall, the article argues that even over an extended period, the electricity connection rate in electrified rural areas remains relatively low, as does electricity usage, whether observed by consumption data or the adoption of new electrical appliances.

The research strategy employed in the study is commendable: the use of household data over time and the comparison of survey data with administrative records enhance the robustness of the findings. The topic is also highly relevant, as the developmental potential of electrification, in the long run, is often assumed rather than empirically demonstrated in the African context. The Rwandan case study is particularly pertinent, as Rwanda is frequently presented as a success story

with significant developmental potential and a track record of rapid electrification.

However, the article still suffers from several weaknesses that need to be addressed before publication. My review focuses primarily on the substance of the argument rather than the quantitative methodology, as my expertise lies in qualitative research and the Rwandan context.

The first issue concerns the robustness of the argument. The article's main claim is that household connection rates in electrified areas remain relatively low. The abstract states that "in connected communities, almost half of the households remain unconnected." However, the data do not unequivocally substantiate this claim. The authors should clarify their data analysis to ensure consistency. For instance, Figure 1 indicates an 82% household connection rate after 10 years. This could be interpreted more positively: having 8 out of 10 households connected to electricity—a service that is not necessarily affordable in one of the world's poorest countries—is actually a significant achievement. Moreover, the authors acknowledge that this figure might be underestimated. In this case, it seems that electrification does lead to relatively rapid connection of households.

The article's claim appears to be based primarily on the observation that about half of the households in a connected community remain unconnected. While this finding is striking, the choice of measurement warrants scrutiny. Measuring connection rates at the "community level" may not be the most appropriate approach, as the authors' definition of "community" does not correspond to standardized units (such as imidugudu, cells, or sectors). Rwanda's hilly terrain means that some areas within a "community" may be notably difficult to access. A more precise and objective definition of "community" is necessary, along with a justification for using this measurement instead of only assessing connection rates within a connection corridor. Defining a community as an aggregate of umudugudu with a portion of it crossed by a low-voltage line is not enough to call the community electrified.

This issue requires clarification because the key argument of the article hinges on it. If 82% of households with access to electricity do connect to the grid, then the overall connection rate is not as low as suggested. If this is the case, the authors might consider reframing the argument in a less striking manner to emphasize the dynamics of electricity connection over time—households that will connect tend to do so early in the process. This findings is particularly relevant for donor cost-benefit analyses and does not seem to be sufficiently addressed in the literature.

Second, the discussion could further elaborate on several points. While electrification's impact may be limited at the individual household level, it may be more significant at the communal level. Health centers, schools, and cooperatives in electrified corridors benefit from electricity access and often connect to the grid. Even if households and small enterprises do not fully capitalize on this potential, as the authors show, key institutions crucial to people's well-being and development likely do. This should be acknowledged and analyzed in light of the study's findings.

Third, an argument worth considering—or at least further emphasizing—is that community electrification is a necessary but insufficient condition for development. The issue is not that electrification often fails cost-benefit analyses (as the article appears to suggest), but rather that existing cost-benefit analyses often fail to incorporate a sufficiently broad range of factors. For example, agricultural price shocks in favor of farmers (as seen in Fetter & Usmani, 2024, in India) could significantly influence electricity adoption and usage in Rwanda. The argument could, therefore, be that electrification should not only be justified on a rights-based basis (as the article argues) but should also be assessed through a more comprehensive economic lens that considers the potential for economic activity expansion before electrification.

Another relevant angle would be to embed this argument within the historical role of electrification in development. Historically, rural electrification followed industrial takeoff, rather than preceding it. Electrification in the countryside may divert precious resources and foreign currency reserves. This perspective would strengthen the argument of the paper about the relatively limited impact of rural electrification for development. For further discussion on this, see Chapter 2 of Robertson, C. (2022). *The Time-Travelling Economist: Why Education, Electricity and Fertility Are Key to Escaping Poverty*. Palgrave Macmillan.

Regarding the discussion, it would be valuable to include the authors' views—or at least hypotheses—on the observed patterns of household connection to the grid. In particular, a discussion of electricity tariffs in Rwanda is warranted, given that electricity prices are reportedly high compared to the rest of Africa. Additionally, situating Rwanda's case within the broader literature on the determinants of low grid connection rates across the continent would enhance the study's contribution.

Finally, the article states that the sample focuses on rural areas, but it would be useful to define what constitutes "rural" in the Rwandan context. Given the country's high population density and rapid urbanization, the rural-urban divide is increasingly blurred, and this should be taken into account in the analysis.

(Remarks on code availability)

Reviewer #3

(Remarks to the Author)

1. Originality of approach or insights.

The paper revisits the ideas of justification and impacts of rural electrification in SSA – with its novelty relying on the revelation of what happens in the long-term. It provided clear pieces of empirical evidence that shows the distributional effects of rural electrification over the long-term, highlighting likely trajectories along the way.

In this context, it provides crucial and innovative insight for policy-makers and funders –that the narrative that rural electrification has significant impacts, at least in the near long-term, needs to be treated with caution! For me, it is this clear insight, and its potential application, that merits the paper's publication. The paper can ignite a deeper discussion or more radical approach to rural electrification in SSA – within a new set of rationale, expectations and limitations – including new opportunities for instrumentalising rural electrification for industrial and societal development.

2. Validity of the work reported.

There is a robust methodology, underpinned by clear explanation and justification of data collection which is broad; followed by approaches to the analysis. These are clearly explained and transparent; and the findings can be seen to derive from a carefully designed data collection approach and subsequent analysis. The scope of the data, in terms of years and sources, has been carefully explained, in a way that justifies the findings. The interpretation of the results is valid and without over-reach. However, it would help the reader if the authors clarified on how the random walk was done.

3. Quality. Brevity and clarity of presentation.

The paper has a clear logical flow and reads well: an introduction and setting of context is accomplished using relevant literature; justification for the paper is also clear. The gap in knowledge is made clear from the synthesis of state of art in the topic in SSA.

4. Significance, relevance and timeliness of the topic.

This is a very relevant issue worthy of dissemination to policy makers, economic development planners, and funders in SSA. The resources expended, expectations, and contingent socio-economic developments premised on rural electrification, make the paper's insight compelling.

I hope this leads to a deeper debate on how the approaches, opportunities and limitations inherent in SSA's rural electrification, and how cost-effectiveness can be viewed within it. The paper is alive to these issues and does well to highlight them in the discussion. can benefit from a paragraph on how results / experience with NG-CBEA differs, and adds value, compared to CBEA which it is designed to improve. The authors need to highlight this for the reader.

5. Adequacy of references.

Up to date sources and relevant sources are appropriately cited.

(Remarks on code availability)

Version 1:

Reviewer comments:

Reviewer #1

(Remarks to the Author)

The authors have sufficiently addressed the queries from my side. The article reads well and is recommended for publication.

(Remarks on code availability)

Reviewer #2

(Remarks to the Author)

The revised version of the paper is clearer and adequately addresses my comments. I think the method section should be higher up in the article but it is only a preference.

I recommend that the article be published.

(Remarks on code availability)

Replies for
Adoption of Electricity in Rural Rwanda 10 Years after
Connection

Nature Communications manuscript NCOMMS-24-84893

*Lise Masselus, Jörg Ankel-Peters, Gabriel Gonzalez Sutil, Vijay
Modi, Joel Mugenyi, Anicet Munyehirwe, Nathan Williams &
Maximiliane Sievert*

July 2025

Below, we address each of the points raised by the referees and summarize how we incorporated these comments in the revised manuscript with color highlighting. Referee comments are shown below in gray boxes. Our responses are below in plain text.

Reviewer 1, comment 1

When the grid is extended to a community, it is considered “**covered**” not “connected” i.e electricity coverage vs electricity access (households connected). Authors should check and rectify this throughout the manuscript. E.g. It is stated in the abstract that “We find that in connected communities, almost half of the households remain unconnected.” This statement could be written as “We find that in covered communities, ...” Please check for similar terminology in other parts of the manuscript—under section 2.2. it is stated that “Despite living in a connected community, not all households connect.”

Thank you for this comment. We agree with the reviewer about the potential confusion in terminology and have changed this throughout the paper. We now also explain the terminology prominently in the introduction:

“Since not all households in a community connect to the grid, throughout the paper we speak of communities being *covered* by the grid and of households being connected or non-connected.”

Reviewer 1, comment 2

In other sub-Saharan African countries where elections are fiercely contested between opposition parties, grid extension is often a strategy for soliciting votes by incumbent parties during elections. In Ghana for example, it is not uncommon to see sign posts along roads in rural areas bearing the inscription “No electricity, no vote.” Hence, the impetus for grid extension may not necessarily be economic-driven or rights-based. But given that Rwanda is more or less a one-party state, this could hardly be the case and as such an economic or right-based impetus would hold true. A generalization of the results to other sub-Saharan Africa countries must take this third political factor into account.

Thank you for this thought.

We absolutely agree that targeting of electrification projects might be used to swing voters (as shown for Ghana in “Briggs, R. C. (2021). Power to which people? Explaining how electrification targets voters across party rotations in Ghana. *World Development*”). We would actually argue that this is a version of a rights-based approach. Because as the referee emphasizes, it is certainly not based on consequentialist economic cost-benefit considerations and rather following a deontological policy approach. In terms of external validity, we therefore also believe that our conclusion holds in such settings in which the “No electricity, no vote” principle applies.

While we believe the subtle differences between the different policy motivations are not of immediate relevance to our paper, the reviewer is right that this is an interesting variety that deserves mentioning. We now do this, very saliently, in the important very last paragraph of our discussion section.

“Many people and also governments in SSA perceive electricity and other infrastructure as a *right* that is derived from normative principles and not from cost-benefit considerations (Ankel-Peters & Schmidt, 2023; Madon et al., 2023; Rao & Min, 2018). This also becomes manifest in countries where politicians use electrification to please (potential) voters (Briggs 2021).”

We thank the reviewer again for putting the spotlight on this important aspect!

Reviewer 1, comment 3
For the surveys in 2022, why was the sample size reduced to 20?

The sample in 2013 was designed to address research questions with higher statistical power requirements (published in Lenz et al., 2017) than the questions addressed in the current paper, such as “how much time do kids spend on studying after nightfall?”. For the present paper, we are interested in more general adoption questions on the community level. We were therefore good with interviewing less households per village in the 2022 survey wave and instead used resources for interviewing community leaders, for example. Also note that given our random walk approach, the reduced sample size does not affect the representativeness of our sample. We have added a clarification on the reduced sample size in the paper (section 4 “Sample selection”).

“A random walk approach was used to select a sample of households representative of all under-grid households. Arriving in a community, the survey team first identified the 50 meters corridor along the distribution lines. They then estimated the total number of households living in this corridor and interviewed every xth household, with x being the total number of households in the corridor divided by the number of interviewed (30 in 2011 and 20 in 2022). A larger sample size in 2013 was required to test hypotheses with higher statistical power requirements (published in Lenz et al. (2017)) than the questions addressed in this paper. The reduced sample size does not affect the representativeness of our sample, as a new random sample was drawn. “

Reviewer 2, comment 1
The first issue concerns the robustness of the argument. The article's main claim is that household connection rates in electrified areas remain relatively low. The abstract states that "in connected communities, almost half of the households remain unconnected." However, the data do not unequivocally substantiate this claim. The authors should clarify their data analysis to ensure consistency. For instance, Figure 1 indicates an 82% household connection rate after 10 years. This could be interpreted more positively: having 8 out of 10 households connected to electricity—a service that is not necessarily affordable in one of the world's poorest countries—is actually a significant achievement. Moreover,

the authors acknowledge that this figure might be underestimated. In this case, it seems that electrification does lead to relatively rapid connection of households.

Thank you for this comment. Indeed, we look at two different populations: households “under the distribution grid”, and the entire community. This is important because for households living directly under the grid, connection fees are much lower. Here, 82% are connected after ten years. However, when including households that live a bit farther away, the connection rate decreases to about half of the households. We believe both findings are very important for the universal access goal. To address the reviewer’s comment, we have added a sentence about the under-grid connection rate to the paper’s abstract and introduction and define more clearly now the two populations throughout the paper.

Abstract:

“we find that nearly half of the households in grid-covered communities remain unconnected. Even among those directly under the distribution grid, electrification rates stagnate slightly above 80%.”

Introduction:

“Using our survey data, we find that around half of households in grid-covered communities remain unconnected. Since distribution lines do not cover the whole community, we also examine connection rates among households living directly under the grid. Here, the electrification rate is higher at around 80%, but has stagnated since 2015.”

The reviewer is right, more than 80% being connected is not bad in such a poor context! The reason we are less positive, though, about the under-grid connection rate is the universal access yardstick. The fact that even not all households connect after several years if connection fees are low poses a serious challenge to the achievement of this universal access goal to which most large donors as well as the Government of Rwanda subscribe.

Regarding the potential underestimation due to our sampling in 2022, note that it is a) very unlikely and b) even if it materializes in some communities, it will not change our verdict about the universal access dilemma. We only mention it to be fully transparent.

We explain this more clearly now in section 2 “Household adoption”.

Section 2:

“Despite living in a covered community, not all households connect. In 2022, up to ten years after community coverage, 82% of the under-grid households are connected. At the entire community level, so including households that live further away from the grid, only 51% are connected. Both observations are important from the universal access goal perspective. Especially in Rwanda’s hilly terrain, extending the grid to the community center does not naturally imply that all parts of the community are within the reach of the distribution lines. But also the fact that the under-grid connection rate has hardly increased since our 3.5-year follow-up is very important. Those households who cannot afford the connection in the short term do not seem to save up money for a later connection. There is one unlikely but theoretically possible caveat: Between 2011 and 2022 REG might have extended distribution lines in some communities, and thus some of the 2022 under-grid households have not been

under the grid for ten years. Among 2011 under-grid households the connection rate would then be higher than 82%. We cannot rule out that new distribution lines were added in some communities, but if it had happened systematically it would have emerged in our community chief interviews (which it did not).”

Reviewer 2, comment 2

The article's claim appears to be based primarily on the observation that about half of the households in a connected community remain unconnected. While this finding is striking, the choice of measurement warrants scrutiny. Measuring connection rates at the "community level" may not be the most appropriate approach, as the authors' definition of "community" does not correspond to standardized units (such as imidugudu, cells, or sectors). Rwanda's hilly terrain means that some areas within a "community" may be notably difficult to access. A more precise and objective definition of "community" is necessary, along with a justification for using this measurement instead of only assessing connection rates within a connection corridor. Defining a community as an aggregate of umudugudu with a portion of it crossed by a low-voltage line is not enough to call the community electrified.

Many thanks for this comment. We are happy to read the referee is familiar with the challenging Rwandan survey context in terms of “what is a community?”. We are fully aware of the difference between administrative units and factual agglomerations. While we adhered to the administrative units when selecting interview partners (we interviewed umudugudu chiefs), by “community” we refer to the factual agglomeration. Which we believe is the relevant definition for infrastructure projects.

Standardized administrative units in Rwanda are not helpful for thinking about electrification projects, because low-voltage lines are most commonly constructed in small centers. These centers are what we called factual agglomeration above, and they are often located at the intersection of several imidugudu (the lowest administrative unit). Commonly, one side of the road would be one umudugudu and the opposite side another one. Typically two to four imidugudu intersect in the center. This is what we call “community”. The next higher administrative level – the cell – bundles around seven imidugudu on average, which are in most cases not one small agglomeration as a center. Cell is therefore too coarse as a sampling cluster. This is why we chose to cluster several imidugudu into “communities”, based on the situation on the ground.

As explained above (and stated more clearly now in the paper), we look at two populations: the under-grid households and the entire (factual) community. The under-grid households are exactly the population that the referee suggests looking at “within a connection corridor”. So we agree with the referee that this is a very relevant population.

Indeed, it is correct that some areas further away from the center are more difficult to access. However, households living in these areas are only included in our definition of the community if they belong to the respective umudugudu intersecting the center and would hence also be included when calculating for example umudugudu connection rates. And this

is because of good reason: In most cases, the nearest access point to the electricity grid for these households is the center that we use for community definition. It is hence correct to consider these households if the aim is universal access to electricity.

We define more clearly now the two populations throughout the whole paper and have added the following Supplementary Note 2.

“We define a community as a group of households clustered around basic infrastructure. We have to resort to this definition of communities, since standardized administrative units in Rwanda are not helpful for thinking about electrification projects. Low-voltage lines are most commonly constructed in small centers. These centers are often located at the intersection of several imidugudu, the lowest administrative unit. Typically, one side of the road belongs to one umudugudu and the opposite side to a different one. Sometimes, three or four imidugudu intersect in the center. The next higher administrative level (cell) bundles around seven imidugudu on average and is therefore too coarse. This is why we chose to cluster several imidugudu together into what we call “communities”, based on the situation on the ground. For all households within such a community, the center used for community definition is usually the nearest access point to the electricity grid.”

We initially had this as a footnote in Section 2 but moved it to the supplementary material because no footnotes are accepted. However, if the reviewer prefers this in the main text, we are happy to do so.

Reviewer 2, comment 3

This issue requires clarification because the key argument of the article hinges on it. If 82% of households with access to electricity do connect to the grid”, then the overall connection rate is not as low as suggested. If this is the case, the authors might consider reframing the argument in a less striking manner to emphasize the dynamics of electricity connection over time—households that will connect tend to do so early in the process. This finding is particularly relevant for donor cost-benefit analyses and does not seem to be sufficiently addressed in the literature.

This comment is related to the two comments above and we are now much clearer in defining the two populations, the under-grid households and the entire community. Both populations are important for policy. The under-grid households can connect at the lowest fee. The fact that not even all these households connect after ten years, challenges the universal access goal. Looking at the entire community is also important because this way, we expose the need to extend distribution lines further (or address this population’s electricity needs via other sources). We see that connecting community centers to the grid is far from sufficient to reach the universal access goal.

For the under-grid households the observation is correct that households connect early on, and that the connection rate does not increase much further. We have added this and now say the following in the paper (section 2):

“In 2022, up to ten years after community coverage, 82% of the under-grid households are connected. At the entire community level, so including households that live further away from the grid, only 51% are connected. Both observations are important from the universal access goal perspective. Especially in Rwanda’s hilly terrain, extending the grid to the community center does not naturally imply that all parts of the community are within the reach of the distribution lines. But also the fact that the under-grid connection rate has hardly increased since our 3.5-year follow-up is very important. Those households who cannot afford the connection in the short term do not seem to save up money for a later connection.”

Regarding the entire community’s connection rate, we cannot say anything about the dynamics, since we only collected this information in 2022. We state this explicitly now and removed the dashed line representing the community connection rate dynamic in Figure 1A, such that it does not appear to be a linear process of connection. Thank you for this remark!

Reviewer 2, comment 4

Second, the discussion could further elaborate on several points. While electrification's impact may be limited at the individual household level, it may be more significant at the communal level. Health centers, schools, and cooperatives in electrified corridors benefit from electricity access and often connect to the grid. Even if households and small enterprises do not fully capitalize on this potential, as the authors show, key institutions crucial to people's well-being and development likely do. This should be acknowledged and analyzed in light of the study’s findings.

Thank you for this suggestion. The referee is right that the focus of the paper is on households and enterprises. But in response to this comment, we have now added information on adoption in schools and health centers, and expectable impacts from those. The information is based on our community leader survey. We asked them about electricity connection, appliance usage in schools and health centers. We elude that the same appliances are being used as in 2013, but otherwise nothing particular happened. Therefore, the positive short-term impact on health centers and schools documented by Lenz et al. (2017) are likely to persist. These impacts are important – but certainly not transformational, even when one would also include benefits like public health and education in the cost-benefit calculation. We now write the following:

“We also asked community leaders about connection, appliance usage and noteworthy uptake of electricity at health centers and schools (although it was not the focus of our survey). First, it is important to observe that 30% of our communities do not have any schools or health infrastructure. Second, all fourteen secondary schools are connected, 21 of the 26 primary schools (81%) and ten of eleven health centers (91%). We furthermore understand that no new uses of electricity have emerged since 2013. The positive short-term results documented by Lenz et al. (2017) regarding schools and health centers are energy cost savings, easier recruitment of skilled staff, improved administrative processes and the addition of some informatics classes for schools. These impacts are likely to sustain, also ten years after electrification – but they are unlikely to be transformative with respect to poverty alleviation (Ankel-Peters et al., 2025a).“

Reviewer 2, comment 5

Third, an argument worth considering—or at least further emphasizing—is that community electrification is a necessary but insufficient condition for development. The issue is not that electrification often fails cost-benefit analyses (as the article appears to suggest), but rather that existing cost-benefit analyses often fail to incorporate a sufficiently broad range of factors. For example, agricultural price shocks in favor of farmers (as seen in Fetter & Usmani, 2024, in India) could significantly influence electricity adoption and usage in Rwanda. The argument could, therefore, be that electrification should not only be justified on a rights-based basis (as the article argues) but should also be assessed through a more comprehensive economic lens that considers the potential for economic activity expansion before electrification.

Thank you for this comment and we largely agree with it. In the conclusion, we do now discuss in more detail the potential of electrification for development, in the face of an external growth stimulus. Yet, while this does not rule out the fact that electricity may help in some place, it does not justify having electricity everywhere. We have moved the discussion from a footnote to the main text and have expanded it.

“There is some evidence that an exogenous stimulus can accelerate growth in combination with electrification. Fetter & Usmani (2024) observe that electrification led to economic development in communities in northwestern India that were simultaneously exposed to a positive exogenous price shock for agricultural products, but not in communities without this shock. While this demonstrates that electricity can enable development in certain places, it does not justify supplying electricity everywhere. The question hence arises whether such potential benefits, either in the far-away future or hinging on exogenous positive shocks, justify today’s high investment and opportunity costs. “

Reviewer 2, comment 6

Another relevant angle would be to embed this argument within the historical role of electrification in development. Historically, rural electrification followed industrial takeoff, rather than preceding it. Electrification in the countryside may divert precious resources and foreign currency reserves. This perspective would strengthen the argument of the paper about the relatively limited impact of rural electrification for development. For further discussion on this, see Chapter 2 of Robertson, C. (2022). *The Time-Travelling Economist: Why Education, Electricity and Fertility Are Key to Escaping Poverty*. Palgrave Macmillan.

Thank you very much for this comment. We agree that this is an interesting perspective on rural electrification in SSA. We have added the following paragraph:

“Alternatively, policy could target existing productive users in regions not covered by the grid. This would be a version of the historical approach in today’s high-income countries where rural electrification typically followed rather than preceded industrial take-off (Robertson, 2022).”

Reviewer 2, comment 7

Regarding the discussion, it would be valuable to include the authors' views—or at least hypotheses—on the observed patterns of household connection to the grid. In particular, a discussion of electricity tariffs in Rwanda is warranted, given that electricity prices are reportedly high compared to the rest of Africa. Additionally, situating Rwanda's case within the broader literature on the determinants of low grid connection rates across the continent would enhance the study's contribution.

Thank you for this comment. In Section 2 "Generalizability across Rwanda and Sub-Saharan Africa" we discuss the external validity of our results and compare connection rates, consumption and appliance usage across Sub-Saharan Africa. This section argues that the results from Rwanda are well in-line with experiences from other SSA country. We show that the lifeline tariff in Rwanda, which is the relevant tariffs for the low consumption levels that we observe, is placed moderately, compared to other SSA countries. Therefore, we believe that it is unlikely that electricity tariffs drive our results on connection rates, consumption and appliance usage. We have added a paragraph on electricity tariffs across SSA in Section 2.

"Rwanda's lifeline tariff introduced in 2017 of around 0.13 USD/kWh (we use the 2015 exchange rate in order to make the tariff comparable to 2015 tariffs reported by Witte & Foster (2020)) is also positioned moderately within the range of lifeline tariffs in Sub-Saharan Africa (Witte & Foster, 2020). While some countries do have lifeline tariffs as low as 0.01 USD/kWh (for example Ethiopia and Cameroon), many countries charge their lowest consumption customers more than 0.15 USD/kWh (Benin, Burkina Faso, Niger, Nigeria, Uganda). Figure 4 shows that countries with very low or high lifeline tariffs do not stand out in terms of appliance ownership, suggesting that tariffs are not the main driver of appliance ownership. Analysing tariff reductions for the lowest consumption tier in Rwanda, Mugenyi et al. (2024) show that reducing the tariff by almost 50%, increases consumption only by slightly more than 10%. This documents that consumption of electricity is price-elastic, but the small size of the effect suggests that other competing and probably essential needs exist within household budgets."

Reviewer 2, comment 8

Finally, the article states that the sample focuses on rural areas, but it would be useful to define what constitutes "rural" in the Rwandan context. Given the country's high population density and rapid urbanization, the rural-urban divide is increasingly blurred, and this should be taken into account in the analysis.

Thank you for your insightful comment. We appreciate your observation regarding the need to clarify the definition of "rural" within the Rwandan context.

In Section 4 "Sample selection", we have included detailed characteristics of the communities to provide a clearer picture of what we classify as rural. For instance, the average community consists of approximately 300 households, with the majority of the population relying on farming as their primary source of income. Prior to grid connection, only a few communities

had a business center or notable entrepreneurial activities. Accessibility is also a key factor; most communities are reachable only via dirt roads, with just one community accessible by an asphalted road. Additionally, public facilities are limited, often restricted to a primary school.

Given these characteristics, these communities align with what are typically considered rural standards in Rwanda. Regarding external validity, we acknowledge the complexity introduced by Rwanda's high population density and rapid urbanization. If these communities are indeed less rural compared to those in other countries, as you suggest, Rwanda could represent an upper bound in terms of the potential benefits of electrification.

We added in Section 4:

“In comparison to other rural areas in SSA, rural Rwanda is relatively densely populated and could therefore represent an upper bound in terms of the potential benefits of rural electrification in SSA.”

Reviewer 3, comment 1

It would help the reader if the authors clarified on how the random walk was done.

Thank you for this comment. We added the clarification on the random walk approach in section 4 “Sample selection”.

“A random walk approach was used to select households representative of all under-grid households. Arriving in a community, the survey team first identified the 50 meters corridor on both sides of the distribution lines. They then estimated the total number of households living in this corridor and interviewed every x th household, with x being the total number of households in the corridor divided by the number of interviewed (30 in 2011 and 20 in 2022). A larger sample size in 2013 was required to test hypotheses with higher statistical power requirements (published in Lenz et al. (2017)) than the questions addressed in this paper. The reduced sample size does not affect the representativeness of our sample, as a new random sample was drawn.”

Reviewer 3, comment 2

I hope this leads to a deeper debate on how the approaches, opportunities and limitations inherent in SSA's rural electrification, and how cost-effectiveness can be viewed within it. The paper is alive to these issues and does well to highlight them in the discussion. can benefit from a paragraph on how results / experience with NG-CBEA differs, and adds value, compared to CBEA which it is designed to improve. The authors need to highlight this for the reader.

We thank the referee for appreciating our approach and contribution. While we are not experts regarding the concepts of community-based environmental assessments (CBEA) and its improved versions (NG-CBEA), we believe our study is well aligned with the NG-CBEA approach. We added a note on this in the supplementary material (Supplementary Note 3):

Discussion section:

“Meanwhile, there are other reasons beyond economic ones that justify rural electrification. Many people and also governments in SSA perceive electricity and other infrastructure as a right that is derived from normative principles and not from cost-benefit considerations (Ankel-Peters & Schmidt, 2023; Madon et al., 2023; Rao & Min, 2018) (see also Supplementary Note 3).”

Supplementary Note 3:

“The “electricity as a right” perspective aligns with the principles of the Next Generation Community-Based Environmental Assessment (NG-CBEA), which emphasizes the importance of comprehensive assessments, meaningful participation of beneficiaries, and the consideration of normative principles in evaluating development interventions (Biswal et al, 2023). This also means embracing logical and practical interdependencies between effectiveness, efficiency, and fairness.”